# Performance Testing and Evaluation of Drum-Type Stem-Separation Device for Pepper Harvester

**Seo-Yong Shin [1], Myoung-Ho Kim [1,2], Yongjin Cho [1,2,*] and Dae-Cheol Kim [1,2,*]**

[1] Department of Bioindustrial Machinery Engineering, Jeonbuk National University, Jeonju 54896, Korea; ssy9970@naver.com (S.-Y.S.); myoung59@jbnu.ac.kr (M.-H.K.)

[2] Institute for Agricultural Machinery & ICT Convergence, Jeonbuk National University, Jeonju 54896, Korea

*   Correspondence: choyj@jbnu.ac.kr (Y.C.); dckim12@jbnu.ac.kr (D.-C.K.)

**Abstract:** The chili pepper harvester has shown potential problems of low pepper stem separation and a high pepper damage rate. The low pepper stem separation has required additional labor, which consists of separating the pepper and stem after pepper harvesting. To improve the stem separation and sorting function of pepper harvesters, three-shaft and four-shaft drum-type stem-separation devices were manufactured, and performance tests were conducted to assess these devices. In an attempt to reduce the damage rate, a brush was used as the teeth in the drum-type stem-separation device. In the factor test, the rotational speeds of shaft 1(A), shaft 2(B), shaft 3(C), and the conveyor for the three-shaft drum were 0.9, 2.7, 1.3, and 0.5 m/s, respectively. The rotational speed of the four-shaft drum was the same as that of the three-shaft drum except for shaft 4(D), and the rotational speed of this additional D was set to 1.3 m/s, which was the same as that of C. In the non-moving status during the non-picking operation of the pepper harvester, the average stem-separation efficiency (SSE) of the four-shaft drum increased by 1.2%, the average pepper with twig rate (PTR) decreased by 5.9%, and the average damage rate (DR) increased by 3.7% compared to the three-shaft drum. In the moving status during the picking operation of the pepper harvester, the SSE of the four-shaft drum increased by 3.6%, the PTR decreased by 9.1%, and the DR increased by 3.8% compared to the three-shaft drum, so an improvement in the pepper stem-separation capacity was observed.

**Keywords:** chili pepper harvester; drum type separating system; prototype; stem-separation of foreign materials; performance; evaluation; field test

## 1. Introduction

The chili pepper is an important economic crop in Korea that is planted in early May and continuously harvested at least three times from the end of July; however, the production of the chili pepper is declining due to the aging of farm households and the slow mechanization rate. The harvest area and production of the chili pepper in Korea are both decreasing due to the increase in imports of chili peppers. The chili pepper harvest area, production, and self-supply rate have all decreased from 74,471 ha, 193,786 tons, and 89%, respectively, in 2000, to 31,146 ha, 60,076 tons, and 40%, respectively, in 2020 [1]. On the other hand, imports of chili peppers have increased 9% each year over the same period, from 30,000 tons to 123,000 tons [2].

The mechanization of the pepper harvester started in the 1970s, and research on the pepper harvester has been carried out in various countries [3–7]. Researchers have actively studied and manufactured chili harvesters to increase pepper harvest and production, but the current pepper harvesters had a high PTR, thus requiring additional manpower to separate twigs from peppers [8–10]. In agriculture, the cost of harvest labor accounts for up to 50% of the total cost of production. The cost of harvest labor could be reduced up to 10% with mechanization or the use of robotic harvesting [11]. However, harvesting by robot has limitations in use depending on the cultivar and environment and is more suitable for greenhouses and small-scale farms [12–15].

Research on chili pepper varieties for use in mechanical harvesting has been conducted [16]. Recently, to evaluate the suitability for harvest mechanization, previous research was conducted on the nutritional and physiological components of various varieties [17–21] as well as plant spacing, productivity, and quality [22–24] according to cultivation methods.

The self-propelled chili pepper harvester consists of the picking and transfer parts that remove peppers from plants, a separation sorting part that removes small branches and leaves, and a collecting part that contains the peppers [25]. Previous studies were conducted on the separation sorting part using a star wheel and a card cleaner [26–29] as well as using an air blower [26,30] to remove small branches. When bigger particles, such as stemmed chili peppers, flow in from the picking part, the development of additional separating and sorting devices are required. Kim et al. [31] obtained a PTR of 24.2–37.8% in a chili harvester comparison test, and suggested the need for a stem-separation and sorting device to reduce the input of additional manpower for sorting work after the chili harvest.

Studies on stem separation used a rotating roller method and a drum-type threshing method. A study on a rotating roller method conducted by Wolf et al. [32] reported that, as a result of a pair of feeding rollers rotating in opposite directions putting a whole paprika plant, including roots, into the two pairs of picking elements, the number of stem-attached peppers was reduced by 50% compared to that of work without a stem-separation system. Esch et al. [33] reduced the amount of stem-attached cherry peppers by more than 90% by placing stem-attached cherry peppers between roller beds of different sizes. Wolf et al. [34] developed a system for separating stems using three cylindrical squeeze rollers rotating clockwise at the top and eight downstream rollers rotating counterclockwise at the bottom. Seiphepi et al. [35] designed and simulated an automated system that detects the color of bell peppers and separates the pepper from the stem by a roller-driven belt.

A study by Gan-Mor et al. [36] examined a drum-type threshing method that used a porous rotating drum to separate leaves from sorghum and parsley stems, and its SSE and DR were 90% and 10%, respectively. Sudajan et al. [37] comparatively analyzed the drum type, rotational speed, and sunflower supply for sorting sunflowers. The optimal sorting drum was found to be the Rasp bar drum, and when the rotation speed was 700 rpm and the supply was 3000 kg/h, the SSE, DR, and loss were 99%, 1.39%, and 0.36%, respectively. To confirm the efficiency of a tractor-attached small tree-cutting chipper, the fuel consumption rate and productivity were compared for the drum and disk types [38]. As a result of the test, the drum-type chipper reduced fuel consumption by 19% and improved productivity by 8% compared to the disk-type chipper. A cassava harvester for stem removal was developed by Jyoti et al. [39]. This machine installed a blade on the outside of the drum to find the optimal angle and thickness of the blade. The best result was obtained at a shear angle of 20°, an approach angle of 30°, and a thickness of 6 mm.

Choi [40] developed a device for separating peppers from stems by depositing harvested pepper stems into a drum-type threshing machine. Pepper SSE and DR were 82.7% and 4.4%, respectively, and the smaller the pepper, the lower the pepper SSE and DR. Funk et al. [25] manufactured and tested a sorting device using a card cleaner and a cylindrical coil drum, and 60–75% of the debris was removed as a result, but the PTR was about 10%, which means additional work was needed to separate stems from peppers. Nam et al. [41] attempted to determine the optimal drum rotation speed, the space between drums, and the drum rotation speed ratio for three rotating drums with steel-bar teeth. They found that the optimal conditions were a drum rotation speed of 70 rpm, a 5 mm space between drums, and a drum rotation speed ratio of 7:3:5, and the pepper SSE and DR were 60% and 15%, respectively. The method suggested in the current reference has low pepper SSE and high DR, so it was necessary to develop a drum-type pepper stem-separation device to improve SSE and DR.

The purpose of this study was to design and manufacture a pepper-stem-separation device for a pepper harvester, and to evaluate its performance. The detailed objectives were as follows.

1.  To design and manufacture three-shaft and four-shaft drum stem-separation devices.
2.  To find the optimal drum rotation speed.
3.  To conduct a comparative performance test and evaluation of the three-shaft and four-shaft drum stem-separation devices.

## 2. Materials and Methods

### 2.1. Chili Pepper Harvester

The pepper harvester picked peppers from the stem by rotating the open helix of the picking parts in the opposite direction while the pepper harvester was moving. The harvested peppers and debris were placed on a conveyor, run through a stem-separation device that separates the peppers from stems, and then stored in the final collection tank after removing debris with an air-wash. The mid-size pepper harvester was composed of a travelling part, a stem-separation part, a debris-sorting part, and a collecting part. The size of length, width, and height of the pepper harvester was 4380 mm, 1850 mm, and 1780 mm, respectively, and its weight was 2020 kg. The pepper harvester used in this study is shown in Figure 1, and the detailed specifications are listed in Table 1.

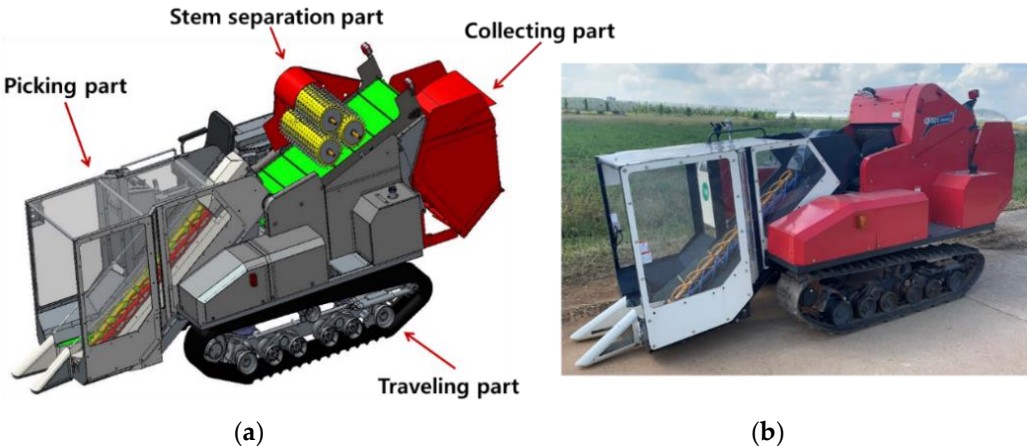

|        (a)         |        (b)         |
|:------------------:|:------------------:|

**Figure 1.** 3D design (**a**) and picture (**b**) of the prototype pepper harvester.

**Table 1.** Specifications of the mid-size chili pepper harvester.

| Item | Data |
|:---:|:---:|
| Power (kW) | 22.5 |
| Weight (kg) | 2020 |
| Standard working speed (m/s) | 0.1–0.3 |
| Rated revolution per minute (rpm) | 2600 |
| Rotational speed of picking part helix (rpm) | 650 |
| Separating method | Drum type and air fan |
| Collecting tank (kg) | 220 |

The drum-type stem-separation device was located at the upper part of a conveyor, and it was driven by deceleration and acceleration according to the gear ratio in a belt power transmission method connected to the engine; the compositions of the 3-shaft drum and the 4-shaft drum are shown in Figure 2. Peppers and stems were separated by the friction of the brush as the stem-attached peppers passed between the drum, with a number of cylindrical brush teeth attached in the circumferential direction and the drum rotating in the opposite directions. According to the order of conveyor transfer, the first drum was drum shaft 1(A), and it was followed in order by shaft 2(B), shaft 3(C), and shaft 4(D).

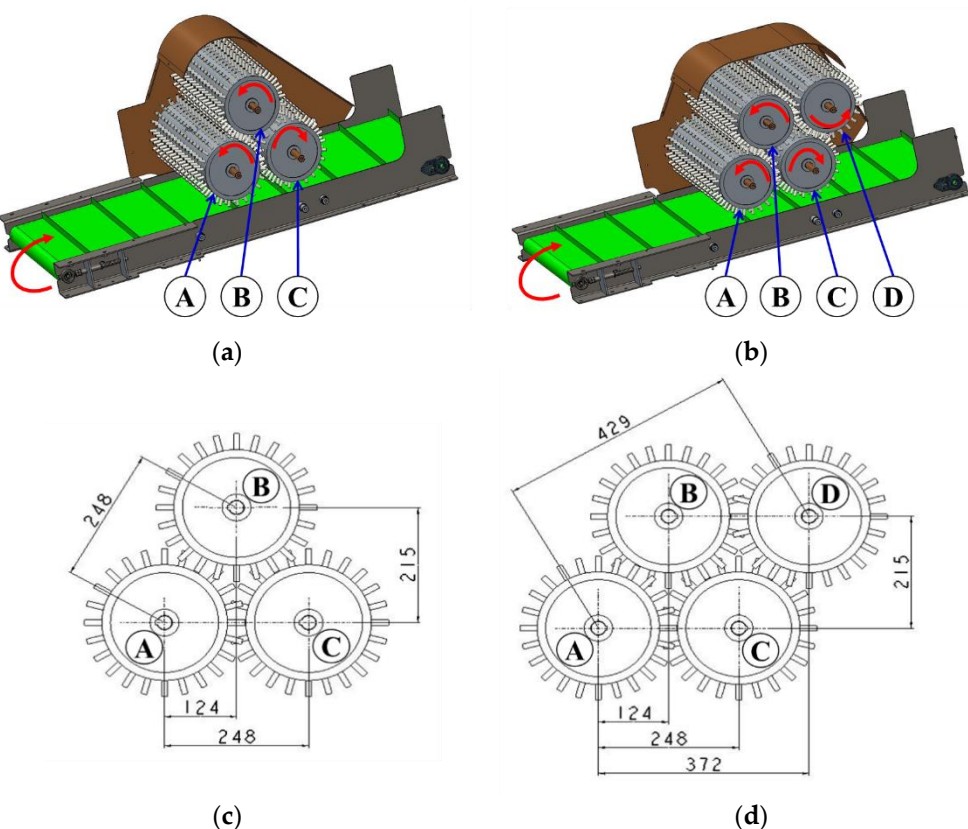

**Figure 2.** 3D design of three-shaft drum (**a**) and four-shaft drum (**b**) of the drum-type stem-separating part. (**c**) Side view of the three-shaft drum type and (**d**) four-shaft drum type.

Figure 3 shows the detailed dimensions of the stem-separation rotating drum and brush teeth. The distances between the centers of rotating drum A and the final drum (C or D) were 248 mm and 429 mm for 3-shaft and 4-shaft, respectively. The distance between the two drums was set to 5 mm so that the teeth brushes did not interfere with each other during rotation.

*2.2. Test Sample*

In this study, to evaluate the performance of both stem-separation devices, two species of pepper—AR Legend and Jeokyoung—were selected because they are suitable for mechanization. AR Legend is large and pigment-rich, has a high fruit yield, and is characterized by a strong resistance to anthrax. Meanwhile, Jeokyoung is a species fostered in the National Institute of Horticultural and Herbal Science. It is a species that is large in size and pigment-rich, and has a high single-harvest rate with a high proportion of red peppers. On 6 May 2019, peppers were planted in the NIHHS open field, and the stem-separation performance test was conducted on 23 August 2019, which was 128 days after planting. Figure 4 shows the pepper field (a) and the physical properties (b) of the pepper used in the test, and Table 2 details the physical properties of the pepper by variety.

**A, D Drum**

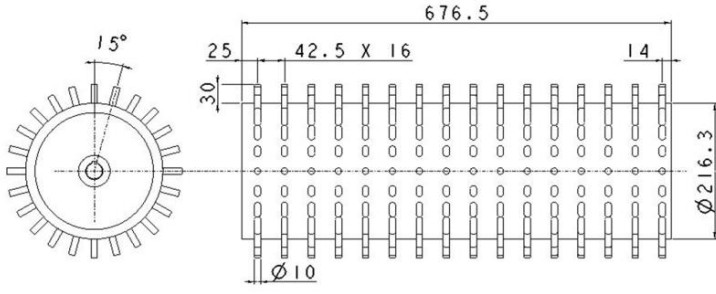

**B Drum**

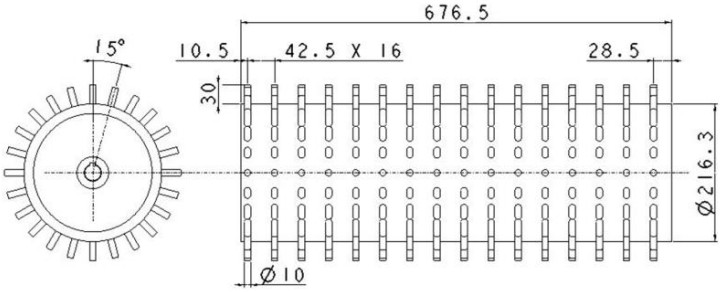

**C Drum**

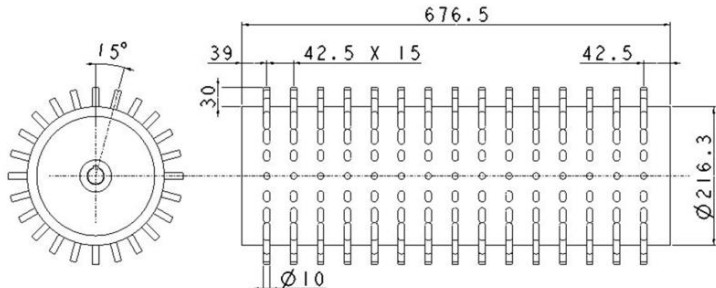

**Figure 3.** Shape and dimensions of each rotating drum and cylindrical brush teeth. A Drum: the first drum installed on shaft 1; B Drum, C Drum, D Drum: the followed drum in order by shaft 2, shaft 3, and shaft 4.

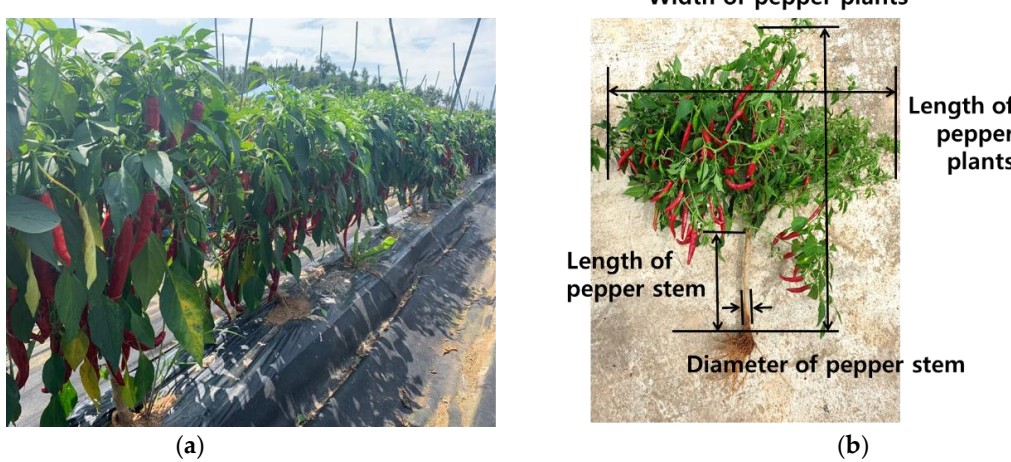

(**a**)                                                                 (**b**)

**Figure 4.** Pictures of the (**a**) pepper plants and (**b**) physical properties of pepper used in this study.

**Table 2.** Average (AVE) and standard deviations (SD) of the physical properties for peppers.

| Variety | | AR Legend | | Jeokyoung | |
|---|---|---|---|---|---|
| Parameter | | AVE | SD | AVE | SD |
| Plant | Length of pepper plant (mm) | 1052.5 | 136.5 | 866.6 | 90.7 |
| | Width of pepper plant (mm) | 814.6 | 111.2 | 776.7 | 125.2 |
| | Length of pepper stem (mm) | 198.3 | 20.3 | 163.3 | 15.3 |
| | Diameter of pepper stem (mm) | 20.5 | 3.8 | 24.6 | 4.1 |
| Fruit | Length (mm) | 118.3 | 13.7 | 112.8 | 20.3 |
| | Maximum diameter (mm) | 23.1 | 2.1 | 22.6 | 2.7 |
| | Weight (g) | 22.4 | 2.3 | 18.7 | 3.2 |
| | Amount (ea/plant) | 88.0 | 6.5 | 96.4 | 5.8 |
| | Detachment force (kgf) | 2.3 | 0.6 | 1.4 | 0.2 |

*2.3. Field Test Method*

To evaluate the performance of the drum stem-separation device, field performance tests were conducted using two species of peppers. Stem-attached pepper samples were prepared for each variety, and five sets of stem-attached peppers were continuously added per each round of testing (Figure 5a). In the non-moving status, during the non-picking operation status of the pepper harvester, pepper samples were placed into the conveyor (Figure 5b). Figure 5c shows the non-moving status during the picking operation status of the pepper harvester, and the pepper samples were placed into the helix driven in the picking operation. Each performance test was repeated three times. In Figure 5d, the field test was conducted by driving at a running speed of 0.2 m/s in the moving status during the picking operation.

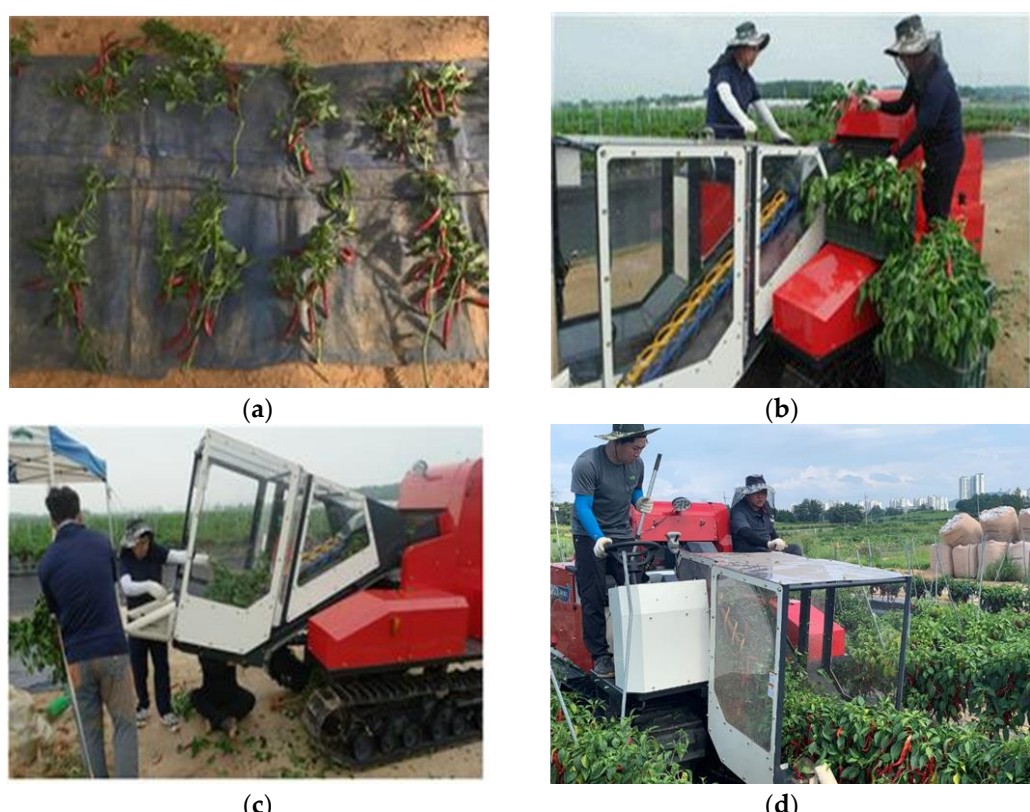

(a)

(b)

(c)

(d)

**Figure 5.** Picture of the (**a**) pepper plants samples, (**b**) non-moving status during non-picking operation, (**c**) non-moving status during picking operation, and (**d**) moving status during picking operation.

The rotation speed ratio of drums A, B, and C of the 3-shaft drum stem-separation device referred to the stem shredder of the vegetable harvester [42]. The A:B:C drum rotation speed ratio of the stem shredder was 1:2:3, and the drum teeth were rotated in opposite directions to shred the stems. As listed in Table 3, the weights of the stem-separated peppers, stem-attached peppers, and damaged peppers were measured at level 6, wherein the rotational speeds of drums B and C were increased based on that of drum A. The D drum speed of the 4-shaft drum device was set to the same rotation speed as the C drum in the 3-shaft drum.

**Table 3.** Factor levels of drum type stem separation.

| | Level Speed (m/s) | | | | | |
|---|---|---|---|---|---|---|
| **Type** | **1** | **2** | **3** | **4** | **5** | **6** |
| A (1st shaft) | 0.3 | 0.4 | 0.4 | 0.5 | 0.6 | 0.9 |
| B (2nd shaft) | 0.5 | 0.7 | 0.8 | 0.9 | 1.8 | 2.7 |
| C (3rd shaft) | 0.8 | 1.1 | 1.3 | 1.3 | 1.3 | 1.3 |
| D (4th shaft) | - | - | - | - | - | 1.3 |
| Conveyor | 0.8 | 0.6 | 0.5 | 0.6 | 0.5 | 0.5 |

### 2.4. Analysis Method

Kim et al. [31] and Choi et al. [40] measured the weights of harvested peppers, peppers without twigs, stem-attached peppers, damaged peppers, unharvested peppers, and damaged peppers. The performance of each pepper harvester was evaluated based on these measured weights. In this study, the SSE, PTR, DR, and ground fall losses ratio (GLR) were calculated by measuring the weights of the various peppers mentioned as in Equations (1)–(4).

SSE is the weight percentage of peppers successfully separated from stems to all peppers (1). PTR was the weight percentage of peppers with twigs after sorting all the peppers (2). DR was the weight percentage of the damaged peppers to all peppers (3). GLR was the weight percentage of the peppers that fell on the ground to all peppers (4).

$$D_{sse} = \frac{P_{ssp}}{P_{ap}} \times 100 \tag{1}$$

where $D_{sse}$ = stem-separation efficiency, %; $P_{ssp}$ = weight of stem-separation pepper fruits, g; and $P_{ap}$ = weight of all pepper fruits, g.

$$D_{ptr} = \frac{P_{pts}}{P_{ap}} \times 100 \tag{2}$$

where $D_{ptr}$ = pepper with twig rate, %; $P_{pts}$ = weight of pepper fruits with twigs after sorting, g; and $P_{ap}$ = weight of all pepper fruits, g.

$$D_{dr} = \frac{P_{dp}}{P_{ap}} \times 100 \tag{3}$$

where $D_{dr}$ = damage rate, %; $P_{dp}$ = weight of damaged pepper fruits, g; and $P_{ap}$ = weight of all pepper fruits, g.

$$D_{glr} = \frac{P_{fp}}{P_{tp}} \times 100 \tag{4}$$

where $D_{glr}$ = ground fall losses ratio, %; $P_{fp}$ = quantity of pepper fruit falling on the ground, g; and $P_{tp}$ = quantity of all pepper fruit, g.

## 3. Results and Discussion

Table 4 presents the performance test results for each factor level according to the speed of the three-shaft drum stem-separation device. When operating from factor level 1 to factor level 6 using the three-shaft drum, the SSE of the AR legend increased, and the PTR and DR of the AR legend decreased. At factor level 6, the average SSE was 76.8%, the average PTR was 20.9%, and the average DR was 2.3%, thus showing the highest efficiency. In this study, to select the optimal drum rotation speed, the rotation speeds of A, B, and C using the three-shaft drum with a factor level of 6 were set to 0.9, 2.7, and 1.3 m/s, respectively, as this showed the highest efficiency. In the four-shaft drum, the rotation speeds of A, B, and C were the same as that of the three-shaft drum, and the rotation speed of D was set to 1.3 m/s, which is the same as C. The conveyor speed was 0.5 m/s.

**Table 4.** Experiment results of field test based on drum type and rotational speed level.

| Drum type | Level | AVE/SD | SSE [1] (%) | PTR [2] (%) | DR [3] (%) |
|---|---|---|---|---|---|
| 3-shaft | 1 | AVE [4] | 51.6 | 45.1 | 3.3 |
| | | SD [5] | 2.8 | 2.7 | 0.5 |
| | 2 | AVE | 59.1 | 38.7 | 2.2 |
| | | SD | 3.4 | 3.8 | 1.4 |
| | 3 | AVE | 53.9 | 42.5 | 3.6 |
| | | SD | 6.3 | 7.1 | 0.8 |
| | 4 | AVE | 57.7 | 38.8 | 3.4 |
| | | SD | 6.2 | 4.1 | 2.1 |
| | 5 | AVE | 64.1 | 33.0 | 2.9 |
| | | SD | 7.2 | 7.9 | 0.7 |
| | 6 | AVE | 76.8 | 20.9 | 2.3 |
| | | SD | 8.2 | 8.5 | 1.3 |

[1] SSE: Stem-separation efficiency; [2] PTR: Pepper with twig rate; [3] DR: Pepper with twig rate; [4] AVE: Average; [5] SD: Standard deviations.

Table 5 lists the field performance test results for AR Legend and Jeokyoung. In the non-moving status during non-picking operation, the average SSEs of AR Legend and Jeokyoung with the three-shaft drum were 70.0% and 69.3%, respectively; the average PTRs were 26.9% and 27.7%, respectively; and the average DRs were 3.1% and 3.0%, respectively, which means the results were similar regardless of pepper species. The average SSEs of AR Legend and Jeokyoung with the four-shaft drum were 64.0% and 72.3%, respectively; the average PTRs were 25.5% and 17.8%, respectively; and the average DRs were 10.5% and 9.9%, respectively, which means that the average PTR of Jeokyoung was lower than that of AR Legend. The overall DR with the four-shaft drum was slightly higher compared to that with the three-shaft drum, but the average PTR decreased, thus confirming that the four-shaft drum is more suitable for stem separation.

**Table 5.** Experiment results of field test based on moving operation of drum type and variety of pepper used for each picking part status.

| Operating Status | | Drum Type | Variety | AVE [1]/SD [2] | SSE [3] (%) | PTR [4] (%) | DR [5] (%) | GAR [6] (%) |
|---|---|---|---|---|---|---|---|---|
| Non-picking operation | Non-moving | 3 | AR Legend | AVE | 70.0 | 26.9 | 3.1 | - |
| | | | | SD | 8.2 | 8.5 | 1.3 | - |
| | | | Jeokyoung | AVE | 69.3 | 27.7 | 3.0 | - |
| | | | | SD | 5.4 | 4.8 | 1.1 | - |
| | | 4 | AR Legend | AVE | 64.0 | 25.5 | 10.5 | - |
| | | | | SD | 2.9 | 3.2 | 1.4 | - |
| | | | Jeokyoung | AVE | 72.3 | 17.8 | 9.9 | - |
| | | | | SD | 6.3 | 6.2 | 2.1 | - |

**Table 5.** *Cont.*

| Operating Status | Drum Type | Variety | AVE [1]/SD [2] | SSE [3] (%) | PTR [4] (%) | DR [5] (%) | GAR [6] (%) |
|---|---|---|---|---|---|---|---|
| Picking operation | Non-moving | | | | | | |
| | | 3 | AR Legend | AVE | 86.0 | 9.3 | 4.7 | - |
| | | | | SD | 0.5 | 0.6 | 1.1 | - |
| | | 4 | AR Legend | AVE | 90.1 | 6.2 | 3.7 | - |
| | | | | SD | 0.7 | 2.3 | 1.6 | - |
| | Moving | 3 | AR Legend | | 65.7 | 22.1 | 9.5 | 2.7 |
| | | 4 | AR Legend | | 69.3 | 13.0 | 11.2 | 6.5 |
| Non-picking operation | Non-moving | 3 | AR Legend | AVE | 70.0 | 26.9 | 3.1 | - |
| | | | | SD | 8.2 | 8.5 | 1.3 | - |
| | | | Jeokyoung | AVE | 69.3 | 27.7 | 3.0 | - |
| | | | | SD | 5.4 | 4.8 | 1.1 | - |
| | | 4 | AR Legend | AVE | 64.0 | 25.5 | 10.5 | - |
| | | | | SD | 2.9 | 3.2 | 1.4 | - |
| | | | Jeokyoung | AVE | 72.3 | 17.8 | 9.9 | - |
| | | | | SD | 6.3 | 6.2 | 2.1 | - |
| Picking operation | Non-moving | 3 | AR Legend | AVE | 86.0 | 9.3 | 4.7 | - |
| | | | | SD | 0.5 | 0.6 | 1.1 | - |
| | | 4 | AR Legend | AVE | 90.1 | 6.2 | 3.7 | - |
| | | | | SD | 0.7 | 2.3 | 1.6 | - |
| | Moving | 3 | AR Legend | | 65.7 | 22.1 | 9.5 | 2.7 |
| | | 4 | AR Legend | | 69.3 | 13.0 | 11.2 | 6.5 |

[1] AVE: Average; [2] SD: Standard deviations; [3] SSE: Stem-separation efficiency; [4] PTR: Pepper with twig rate; [5] DR: Pepper with twig rate; [6] GAR: Ground fall losses ratio.

In the non-moving status during picking operation, the average SSEs with the three-shaft drum were 86.0% and 90.1%, respectively; the average PTRs were 9.3% and 6.2%, respectively; and the average DRs were 4.7% and 3.7%, respectively. Compared to the three-shaft drum, the average SSE with the four-shaft drum increased by 4.1%, the average PTR decreased by 3.1%, and the average DR increased by 1%, which means that the four-shaft drum had higher performance than the three-shaft drum. The three-shaft and four-shaft drum-type stem-separation devices used in this study showed higher SSE (86.0%, 90.1%) than that of Nam [41] (63.8%) and Choi [40] (82.7%), and the average DR (4.7%, 3.7%) significantly decreased compared to Nam [41] (12.9%), which was considered to have higher performance.

Table 5 lists the field performance test results in the moving status during picking operation for an 8 m sector using a pepper harvester equipped with the pepper stem-separation device developed in this study. The average performances of the three-shaft drum and four-shaft drum showed SSEs of 65.7% and 69.3%, respectively; average PTRs of 22.1% and 13.0%, respectively; and average DRs of 9.5% and 11.2%, respectively. SSE increased by 3.6%, PTR decreased by 9.1%, and DR increased by 1.7% in the four-shaft drum compared to the three-shaft drum, so the four-shaft drum had better stem-separation device performance than the three-shaft drum. However, further research is needed to determine how to reduce DR.

Table 6 presents the results of a two-way analysis of variance intended to check the effects on SSE, PTR, and DR according to the pepper species and the number of drums. For the statistical analysis of two-way analysis of variance, the commercial software SAS (Version 9.4, SAS Institute, Cary, NC, USA, 2019) was used, and the analysis was performed at a confidence level of 95%. As a result of the variance analysis of PTR and DR in regard to the pepper species and the number of drums, the number of drums was confirmed to affect PTR and DR with P-values of 0.0219 and 0.0001, respectively, which were both smaller than

the significance level of 0.05. In addition, the interaction between the number of drums and pepper species turned out to have no impact on PTR.

**Table 6.** Statistical results of field test based on shafts of drum type and variety of pepper.

| Item | Source | DF | Anova SS | Mean Square | F Value | Pr > F |
|---|---|---|---|---|---|---|
| | Shaft | 1 | 99.4200 | 5.0700 | 0.2800 | 0.6140 |
| | Variety | 1 | 38.8800 | 38.8800 | 2.1500 | 0.1803 |
| SSE [1] | Shaft*variety | 1 | 55.4700 | 55.4700 | 3.0700 | 0.1176 |
| | Error | 8 | 144.3400 | 18.0425 | | |
| ☐ | Total | 11 | 243.7600 | ☐ | ☐ | ☐ |
| | Shaft | 1 | 99.1875 | 99.1875 | 8.06 | 0.0219 |
| | Variety | 1 | 33.6675 | 33.6675 | 2.74 | 0.1367 |
| PTR [2] | Shaft*variety | 1 | 52.5008 | 52.5008 | 4.27 | 0.0728 |
| | Error | 8 | 98.4666 | 12.3083 | | |
| ☐ | Total | 11 | 283.8225 | ☐ | ☐ | ☐ |
| | Shaft | 1 | 148.4033 | 148.4033 | 87.17 | 0.0001 |
| | Variety | 1 | 0.1633 | 0.1633 | 0.1 | 0.7647 |
| DR [3] | Shaft*variety | 1 | 0.03 | 0.03 | 0.02 | 0.8977 |
| | Error | 8 | 13.62 | 1.7025 | | |
| ☐ | Total | 11 | 162.2166 | ☐ | ☐ | ☐ |

[1] SSE: Stem-separation efficiency; [2] PTR: Pepper with twig rate; [3] DR: Pepper with twig rate.

## 4. Conclusions

In this study, to improve the PTR after harvest with the current pepper harvester, two types of stem-separation devices were manufactured: A three-shaft drum and a four-shaft drum. For red pepper varieties, AR Legend and Jeokyoung varieties were used as test samples, and the optimal rotation speed was selected based on factor performance tests for each drum and the conveyor rotation speed. The optimum rotational speeds were 0.9 m/s, 2.7 m/s, 1.3 m/s, and 0.5 m/s for shaft1, shaft2, shaft3, and the conveyor, respectively. To evaluate the comparative performances of the three-shaft drum and four-shaft drum stem-separation devices, the performance was tested according to each state, such as moving or non-moving status during picking operation or not, respectively. The stem-separation device developed in this study showed improved performance compared to the currently existing products. The three-shaft and four-shaft drum-type stem-separation devices used in this study resulted in 86.0% and 90.1% higher SSE than up to 82.7% of the results in previous studies. The four-shaft drum has higher SSE and lower PTR than the three-shaft drum, so the four-shaft drum was confirmed to be more suitable for stem separation. However, for the four-shaft drum, the material cost and processing cost were 23% higher than the three-shaft drum, and it is necessary for manufacturers to choose between the three-shaft drum and the four-shaft drum in consideration of performance and price. The drum-type stem-separation device developed in this study can be supplied to farms to additionally reduce labor costs after harvesting using a machine. The mechanism using the drum-type is used in various industrial applications [38,43] and the drum type is concluded as one of the most effective stem-separation devices for pepper harvesters. To apply the stem-separation device to diverse species of peppers, it is necessary to conduct additional experiments with species other than the species tested in this study.

**Author Contributions:** Conceptualization, S.-Y.S. and Y.C.; performed the experiments and analyzed the data, S.-Y.S. and D.-C.K.; methodology, M.-H.K.; writing—original draft preparation, S.-Y.S. and Y.C.; project administration, Y.C.; writing—review and editing, Y.C., D.-C.K. and M.-H.K. All authors have read and agreed to the published version of the manuscript.

**Funding:** This work was supported by the Korea Institute of Planning and Evaluation for Technology in Food, Agriculture and Forestry (IPET) through the Agriculture, Food and Rural Affairs Convergence Technologies Program for Educating Creative Global Leader Program, funded by the Ministry

of Agriculture, Food and Rural Affairs (MAFRA) (716001-7), and through the Advanced Production Technology Development Program, funded by the Ministry of Agriculture, Food and Rural Affairs (MAFRA) (320018-2).

**Institutional Review Board Statement:** Not applicable.

**Informed Consent Statement:** Not applicable.

**Data Availability Statement:** Not applicable.

**Conflicts of Interest:** The authors declare no conflict of interest.

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
