# Peer review of "Performance Testing and Evaluation of Drum-Type Stem-Separation Device for Pepper Harvester"

_applsci, doi:10.3390/app11199225_

Round 1

Reviewer 1 Report

Comments:

The manuscript entitled “Performance Testing and Evaluation of Drum-type Stem Separation Device for Pepper Harvester” is an interesting work, but the paper needs substantial amendments to be suitable for publication. This article can be accepted after addressing the following issues:

  1. Generally, there is a paragraph at the start of the abstract which highlights the research problems and their significance.
  2. Research significance and novelty are not clear.
  3. Spaces need to be added between numbers and units (e.g., 3,000kg/h, 700rpm, 2,020kg, etc.). By the way, the space between the number and the percent sign can be omitted.
  4. Incorrect citation formate for Choi [9] developed.
  5. Most of the cited references in the introduction are not recent. Is there no work done in recent years?
  6. What does (L X W X H) mean? Additionally, why author mention only X such as 4,380 X.
  7. The sentence should be modified as “various peppers mentioned as in equation (1) – (4).”
  8. The conclusion section should be revised considering the key findings.

Author Response

Dear Reviewers and Editors,

We truly appreciate the editor and reviewers for allowing us to revise our work and have kind comments to improve our manuscript.

Thank you for the good suggestions to improve our manuscript. Below we have copied the text of those suggestions and added our response in italics (with red). We have incorporated manuscript changes based on almost all of these comments. Where we have given line numbers below, they refer to the version of the MS Word document without tracked changes.

Regards,

Yongjin Cho and co-authors

----------------------------------------------------

Response to comments: applsci-1385825

The manuscript entitled “Performance Testing and Evaluation of Drum-type Stem Separation Device for Pepper Harvester” is an interesting work, but the paper needs substantial amendments to be suitable for publication. This article can be accepted after addressing the following issues:

Point 1:

Generally, there is a paragraph at the start of the abstract which highlights the research problems and their significance

Response 1: We added sentences (line 9-11) in the revised manuscript as you advised.

Point 2:  

Research significance and novelty are not clear.

Response 2: We added sentences (line 9-11) in the revised manuscript as you advised.

Point 3:

Spaces need to be added between numbers and units (e.g., 3,000kg/h, 700rpm, 2,020kg, etc.). By the way, the space between the number and the percent sign can be omitted.

Response 3: We added spaces between numbers and units (line 78, 114-116, 168) in the revised manuscript as you advised.

Point 4:

Incorrect citation formate for Choi [9] developed:

Response 4: We revised this format (line 379-380) in the revised manuscript as you advised.

Point 5:

Most of the cited references in the introduction are not recent. Is there no work done in recent years?

Response 5: We cited recent studies in the introduction (line 42-57, 70-72, 79-86) in the revised manuscript as you advised.

Point 6:

What does (L X W X H) mean? Additionally, why author mention only X such as 4,380 X.

Response 6: We revised sentences (line 114-116) in the revised manuscript as you advised.

Point 7:

The sentence should be modified as “various peppers mentioned as in equation (1) – (4).”

Response 7: We revised sentences (line 190-191) in the revised manuscript as you advised.

Point 8:

The conclusion section should be revised considering the key findings.

Response 8: We added sentences (line 282-284, 290-292) in the revised manuscript as you advised.

Reviewer 2 Report

The topic we study is important from the point of view of machine design and energy consumption. However, only 13 entries in references indicate that the authors did not conduct a thorough analysis of the state of the art. In addition, the literature is in most cases older than 10 years. The authors need to correct this. Therefore, major fixes are recommended in the first place. The article must contain at least 30 references not older than 5 years. Depending on the subject, this is the minimum that indicates the quality of the publication and its novelty in the world of science. Abstract - is correct Introduction - needs a lot of improvement as I wrote before. The methodology, Results and Discussion - are carried out reliably, the conclusions are based on the conducted research. At the end of discussion, authors may add the following sentence: Drum mechanisms are used in many industrial applications, the research results confirm that they are one of the most effective in the separation of stalks, which is consistent with the results in other industries, e.g. when shredding branches [https://doi.org/10.1016/j.renene.2021.09.039, https://doi.org/10.14214/sf.930]. 

Author Response

Dear Reviewers and Editors,

We truly appreciate the editor and reviewers for giving us an opportunity to revise our work and have kind comments to improve our manuscript.

Thank you for the good suggestions to improve our manuscript. Below we have copied the text of those suggestions and added our response in italics (with red). We have incorporated manuscript changes based on almost all of these comments. Where we have given line numbers below, they refer to the version of the MS Word document without tracked changes.

Regards,

Yongjin Cho and co-authors

----------------------------------------------------

Response to comments: applsci-1385825

Point 1:

The topic we study is important from the point of view of machine design and energy consumption. However, only 13 entries in references indicate that the authors did not conduct a thorough analysis of the state of the art. In addition, the literature is in most cases older than 10 years. The authors need to correct this. Therefore, major fixes are recommended in the first place. The article must contain at least 30 references not older than 5 years. Depending on the subject, this is the minimum that indicates the quality of the publication and its novelty in the world of science. Abstract - is correct Introduction - needs a lot of improvement as I wrote before. The methodology, Results and Discussion - are carried out reliably, the conclusions are based on the conducted research. At the end of discussion, authors may add the following sentence: Drum mechanisms are used in many industrial applications, the research results confirm that they are one of the most effective in the separation of stalks, which is consistent with the results in other industries, e.g. when shredding branches [https://doi.org/10.1016/j.renene.2021.09.039, https://doi.org/10.14214/sf.930].

 Response 1: Thank you for drawing this to our attention. We cited recent studies in the introduction (line 42-57, 70-72, 79-86) in the revised manuscript as you advised.

In the conclusion part, we added sentences (line 282-284, 290-292) in the revised manuscript as you advised.

Round 2

Reviewer 2 Report

Thank you for the answer you received. However, the statement is not entirely true. "In the conclusion part, we added sentences (line 282-284, 290-292) in the revised manuscript as you advised. " The authors introduced new tasks to the text, but the sentences: "The mechanism using the drum-type is used in various industrial applications and the drum-type is concluded as one of the effective stem separation devices for pepper harvester." They did not support the literature review. Authors here should confirm the application with the cited literature, e.g. "The mechanism using the drum-type is used in various industrial applications [https://doi.org/10.1016/j.renene.2021.09.039, https://doi.org/10.14214/sf.930] and the drum -type is concluded as one of the effective stem separation devices for pepper harvester. "

Author Response

Dear Reviewers and Editors,

We truly appreciate the editor and reviewers for giving us an opportunity to revise our work and have kind comments to improve our manuscript.

Thank you for the good suggestions to improve our manuscript. Below we have copied the text of those suggestions and added our response in italics (with red). We have incorporated manuscript changes based on almost all of these comments. Where we have given line numbers below, they refer to the version of the MS Word document without tracked changes.

Regards,

Yongjin Cho and co-authors

----------------------------------------------------

Response to comments: applsci-1385825

Response 1: Thank you very much for drawing this to our attention. We are missing the literature review. In the conclusion part, we added references to support the literature review (Line 289-290, Line 388-389) in the revised manuscript as you advised.
